# Surveillance of invasive *Aedes* mosquitoes along Swiss traffic axes reveals different dispersal modes for *Aedes albopictus* and *Ae. japonicus*

Pie Müller[1,2]*, Lukas Engeler[3], Laura Vavassori[1,2], Tobias Suter[1,2], Valeria Guidi[3], Martin Gschwind[1,2], Mauro Tonolla[3], Eleonora Flacio[3]

**1** Swiss Tropical and Public Health Institute, Socinstrasse, Basel, Switzerland, **2** University of Basel, Petersplatz, Basel, Switzerland, **3** Laboratory of Applied Microbiology, University of Applied Sciences and Arts of Southern Switzerland, Bellinzona, Switzerland

* pie.mueller@swisstph.ch

**Data Availability Statement:** All relevant data are within the manuscript and its Supporting Information files.

## Abstract

Over the past three decades, Europe has witnessed an increased spread of invasive aedine mosquito species, most notably *Aedes albopictus*, a key vector of chikungunya, dengue and Zika virus. While its distribution in southern Europe is well documented, its dispersal modes across the Alps remain poorly investigated, preventing a projection of future scenarios beyond its current range in order to target mosquito control. To monitor the presence and frequency of invasive *Aedes* mosquitoes across and beyond the Alps we set oviposition and BG-Sentinel traps at potential points of entry with a focus on motorway service areas across Switzerland. We placed the traps from June to September and controlled them for the presence of mosquitoes every other week between 2013 and 2018. Over the six years of surveillance we identified three invasive *Aedes* species, including *Ae. albopictus*, *Ae. japonicus* and *Ae. koreicus*. Based on the frequency and distribution patterns we conclude that *Ae. albopictus* and *Ae. koreicus* are being passively spread primarily along the European route E35 from Italy to Germany, crossing the Alps, while *Ae. japonicus* has been expanding its range from northern Switzerland across the country most likely through active dispersal.

## Author summary

Because of global trade of used tyres and ornamental plants, invasive mosquitoes of the genus *Aedes* are spreading passively between continents. Within continents, adults are frequently travelling along roads as hitchhikers in motorised vehicles and may then colonise new areas. Because some *Aedes* mosquitoes are competent to transmit diseases they threaten public and veterinary health. In Europe, the Asian tiger mosquito, *Aedes albopictus* is of particular concern as it is a vector of chikungunya, dengue and Zika virus. While its distribution in southern Europe is well documented, its dispersal modes across the Alps remain poorly investigated, preventing a projection of future scenarios beyond its current range in order to target mosquito control. To monitor the introduction of invasive

**Funding:** This work received funding to PM, LE, LV, TS, VG, MG, MT and EF from the Swiss Federal Office for the Environment FOEN under the contract numbers 00.0303.PZ/M235-1640 and 00.0303.PZ/Q224-1811, and the pilot programme "Adaptation to climate change". The funder had no role in study design, data collection and analysis, decision to publish, or preparation of the manuscript.

**Competing interests:** The authors have declared that no competing interests exist.

*Aedes* mosquitoes beyond the Alps we placed traps at motorway service areas across Switzerland. Between 2013 and 2018 we identified three invasive *Aedes* species, including *Ae. albopictus*, *Ae. koreicus* (Korean bush mosquito) and *Ae. japonicus* (Japanese bush mosquito). Based on the frequency and distribution patterns we conclude that *Ae. albopictus* and *Ae. koreicus* are being passively spread primarily along the European route E35 from Italy to Germany, while *Ae. japonicus* has been expanding its range across Switzerland mainly through active dispersal.

## Introduction

In the wake of globalisation and environmental change invasive *Aedes* mosquito species are an emerging public health threat [1, 2]. Several *Aedes* species are competent disease vectors and have a particularly high invasive potential as they adapt to new environments and produce eggs that can withstand desiccation for several months. Therefore, their eggs are passively displaced across the globe by the international trade of used tyres and ornamental plants. At the same time, international travel is increasing and, as a result, more and more travellers are returning from disease endemic countries with infections [3] that may then be locally transmitted where competent, invasive *Aedes* mosquitoes are present.

Among the *Aedes* mosquitoes, *Aedes aegypti* (Linnaeus, 1762), the yellow fever mosquito and *Ae. albopictus*, the Asian tiger mosquito (Skuse, 1894) are particularly important vectors because they are widespread, well adapted to human habitats and competent to transmit a range of medically important viruses. *Aedes aegypti* is an important vector of yellow fever, dengue, chikungunya and Zika virus and competent for several other arboviruses [4]. *Aedes albopictus* is also a vector of dengue, chikungunya and Zika virus, and has been shown to transmit at least another 23 viruses under experimental conditions, including yellow fever and West Nile virus [5]. In contrast to *Ae. aegypti* that favours a tropical climate, *Ae. albopictus* can produce diapausing eggs allowing the species to persist cold periods that are critical to adult survival. Photoperiodic diapause together with adaptation to human habitats has allowed for its invasion of more temperate regions, including Europe [6]. In Europe, *Ae. albopictus* has become an emerging health threat and has been associated with autochthonous transmissions of chikungunya in Italy [7,8] and France [9–11], dengue in Croatia [12], France [13,14] and Spain [15], and for the first time with Zika in France [16]. In addition to its vector potential, *Ae. albopictus* is primarily perceived as a considerable nuisance mosquito because it affects people when they spend time outdoors during the day [17].

*Aedes albopictus* is considered the most invasive mosquito species worldwide [18]. Within 40 years the species spread from its native range in South-East Asia to America, Europe, Africa, Australia and several islands in the Pacific. Like other invasive *Aedes* species, the eggs of *Ae. albopictus* are primarily spread passively over long distances across continents through the international trade of used tyres and ornamental plants. Within continents, adult mosquitoes are frequently hitch riding in vehicles and subsequently displaced along the roads [19]. In mainland Europe, *Ae. albopictus* was first recorded in Albania in 1979 [20] and later in northern Italy in the early 90's [21,22]. Less than a decade later it was established in the northern and central regions of Italy [23,24] from where it spread further across Europe [25]. Currently, *Ae. albopictus* is established all over the Mediterranean region from Spain to Greece [26]. In Switzerland, *Ae. albopictus* was first detected in the Canton of Ticino at a motorway service area and the Airport Locarno Magadino in 2003 [27], and has since then gradually infested several regions of the Canton south of the Alps [28,29].

In addition to *Ae. albopictus*, two other invasive *Aedes* species, originating from East Asia, have previously been reported from Switzerland; *Ae. japonicus japonicus* (Theobald, 1901), the Asian or Japanese bush mosquito [30], hereafter called *Ae. japonicus*, and *Ae. koreicus* (Edwards, 1917), the Korean bush mosquito. In contrast to *Ae. albopictus*, these two species are generally considered to be less relevant in terms of public health because there is little evidence for disease transmission [1]. Nevertheless, under laboratory conditions, both species show vector competence for a range of human pathogenic viruses [31] and filarial nematodes [32,33]. *Aedes japonicus* and *Ae. koreicus* seem to prefer lower temperatures as compared to *Ae. albopictus* and, therefore, these species show a high invasive potential in the more temperate regions of Central Europe, including many areas across Switzerland [34,35].

In Switzerland *Ae. japonicus* was first identified in 2008 when it had already colonised an area of about 1,400 km$^2$ in the North across the border with southwestern Germany [36]. In Europe, it had previously only been found in smaller areas, including France [37] and Belgium [38], while until now *Ae. japonicus* has already been reported from 12 European countries [39]. *Aedes koreicus*, on the other hand, was first identified in ovitraps near the Swiss-Italian border in 2013 [40]. Before that time *Ae. koreicus* had also been reported from Belgium [41], north-eastern Italy [42] and European Russia [43]. Today, *Ae. koreicus* is present in at least seven European countries, including Austria, Belgium, Germany, Hungary, Italy, Slovenia and Switzerland [30, 40–42, 44–46].

Until 2013 no *Ae. albopictus* had been reported from Switzerland outside the Canton of Ticino. Similarly, the other two invasive *Aedes* species, *Ae. japonicus* and *Ae. koreicus* had shown very localised distribution patterns [36,40]. The Alps potentially constitute a natural physical and climatic barrier for the migration of invasive *Aedes* species from South to North, and vice versa. Alternatively, *Ae. albopictus* and *Ae. koreicus* might have remained unnoticed outside Ticino in the absence of a nationwide surveillance programme.

While its distribution in southern Europe is well documented [26], its dispersal modes across the Alps remain poorly investigated, preventing a projection of future scenarios beyond its current range in order to target mosquito control. To monitor the presence and frequency of invasive *Aedes* mosquitoes beyond the Alps we set oviposition and BG-Sentinel traps at potential points of entry with a focus on motorway service areas across Switzerland. We set the traps between 2013 and 2018 from June to September and controlled them for the presence of mosquitoes every second week. We found both *Ae. albopictus* and *Ae. koreicus* being introduced primarily along the route from Italy to Germany across the Alps, while the pattern for *Ae. japonicus* implies a more active, radial range expansion away from its initial distribution in the North.

## Methods

### Mosquito sampling

From 2013 to 2018 we sampled invasive *Aedes* mosquitoes at potential points of entry (sites) along the major traffic axes in Switzerland, including motorway service areas (n = 35), commercial harbours (n = 3), international airports (n = 2) and the railway station in Chiasso at the Swiss-Italian border (Table 1). We deployed oviposition traps (ovitraps) to collect eggs and BG-Sentinel version 1 traps (Biogents, Regensburg, Germany) to catch host-seeking adults. Each year the traps were set from end of June (i.e. calendar week 26 or 27) to mid-September (i.e. calendar week 36 or 37) as this corresponds to the activity peak of *Ae. albopictus* in northern Italy and in the infested areas of the Canton of Ticino [47]. We controlled the traps for the presence of mosquitoes every other week, leading to six sampling rounds per year.

**Table 1. Sampling sites and number of ovitraps and BG-Sentinel traps per site.**

| Site | Canton | Coordinates (degrees) | Elevation (m.a.s.l.) | OT (n) | BGS (n) |
|---|---|---|---|---|---|
| A1 Bavois-Est | VD | N 46.67460, E 6.56958 | 555 | 3 | - |
| A1 Bavois-Ouest | VD | N 46.67400, E 6.57067 | 555 | 3 | - |
| A1 Deitingen-Nord | SO | N 47.22889, E 7.62275 | 423 | 3 | 1 |
| A1 Deitingen-Süd | SO | N 47.22601, E 7.61578 | 423 | 3 | - |
| A1 Forrenberg-Nord | ZH | N 47.52667, E 8.73433 | 468 | 3 | - |
| A1 Grauholz | BE | N 46.99029, E 7.47769 | 584 | 6 | 1 |
| A1 Gunzgen-Nord | SO | N 47.31012, E 7.83232 | 433 | 3 | - |
| A1 Gunzgen-Süd | SO | N 47.31015, E 7.84734 | 444 | 3 | - |
| A1 Kemptthal | ZH | N 47.44858, E 8.70026 | 503 | 4 | - |
| A1 Kölliken-Nord | AG | N 47.33007, E 8.03098 | 438 | 3 | - |
| A1 Kölliken-Süd | AG | N 47.32289, E 8.02166 | 446 | 3 | - |
| A1 La Côte Jura | VD | N 46.44707, E 6.29995 | 435 | 3 | 1* |
| A1 La Côte Lac | VD | N 46.44462, E 6.29673 | 429 | 3 | 1 |
| A1 Rose de la Broye | FR | N 46.83206, E 6.85950 | 489 | 6 | 1 |
| A1 St. Margrethen-Nord | SG | N 47.46151, E 9.60356 | 399 | 3 | 1 |
| A1 St. Margrethen-Süd | SG | N 47.46066, E 9.60297 | 400 | 3 | 1 |
| A1 Thurauen-Nord | ZH | N 47.46100, E 9.09423 | 509 | 3 | 1 |
| A1 Würenlos-Nord | AG | N 47.43904, E 8.34747 | 392 | 3 | - |
| A1 Würenlos-Süd | AG | N 47.43907, E 8.34616 | 394 | 3 | - |
| A2 Bellinzona-Nord | TI | N 46.20982, E 9.02753 | 238 | 3 | - |
| A2 Bellinzona-Sud | TI | N 46.18211, E 9.00164 | 227 | 3 | 1* |
| A2 Coldrerio | TI | N 45.84970, E 8.98612 | 312 | 3 | - |
| A2 Eggberg | SO | N 47.33595, E 7.82834 | 549 | 3 | - |
| A2 Gotthard-Nord | UR | N 46.84612, E 8.63370 | 457 | 3 | 1* |
| A2 Gotthard-Süd | UR | N 46.84706, E 8.63203 | 457 | 3 | 1 |
| A2 Neuenkirch-Nord | LU | N 47.11365, E 8.23129 | 560 | 3 | 1* |
| A2 Neuenkirch-Süd | LU | N 47.11063, E 8.23380 | 548 | 3 | 1 |
| A2 Pratteln—Nord | BL | N 47.52759, E 7.70125 | 273 | 3 | 1 |
| A2 Pratteln—Süd | BL | N 47.52710, E 7.70055 | 272 | 3 | 1 |
| A2 San Gottardo-Sud | TI | N 46.51521, E 8.66768 | 1015 | 3 | - |
| A2 San Gottardo-Sud Stalvedro | TI | N 46.52080, E 8.63637 | 1064 | 3 | 1 |
| A2 Teufengraben | SO | N 47.33316, E 7.82170 | 522 | 3 | 1 |
| A9 St-Bernard | VS | N 46.12759, E 7.06026 | 455 | 3 | - |
| A13 Heidiland | GR | N 47.01092, E 9.51217 | 501 | 3 | 1* |
| A13 Rheintal-Ost | SG | N 47.14597, E 9.50159 | 455 | 3 | - |
| A13 Rheintal-West | SG | N 47.14622, E 9.49989 | 455 | 3 | - |
| Auhafen | BL | N 47.54023, E 7.66176 | 258 | 6 | 1 |
| Bahnhof Chiasso | TI | N 45.84059, E 9.00212 | 247 | 6 | - |
| Genève Aéroport | GE | N 46.23701, E 6.10910 | 418 | 6 | 1 |
| Flughafen Zürich | ZH | N 47.45399, E 8.57711 | 432 | 6 | 1 |
| Innenhof Swiss TPH | BS | N 47.55564, E 7.57809 | 279 | 3 | 1* |
| Rheinhafen Kleinhüningen–Hafenbecken 1 | BS | N 47.58450, E 7.58855 | 249 | 6 | 1 |
| Rheinhafen Kleinhüningen–Hafenbecken 2 | BS | N 47.58705, E 7.59879 | 253 | 6 | 1 |
| **Total** | | | | **154** | **24** |

Swiss cantons: AG: Aargau; BE: Bern; BL: Basel-Landschaft; BS: Basel-Stadt; FR: Fribourg; GE: Genève; GR: Graubünden; LU: Luzern; SG: St. Gallen: SO: Solothurn; TI: Ticino; UR: Uri; VD: Vaud; VS: Wallis; ZH: Zürich. OT: ovitrap; BGS: BG-Sentinel trap. The star (*) next to the number indicates that the trap was additionally supplied with $CO_2$.

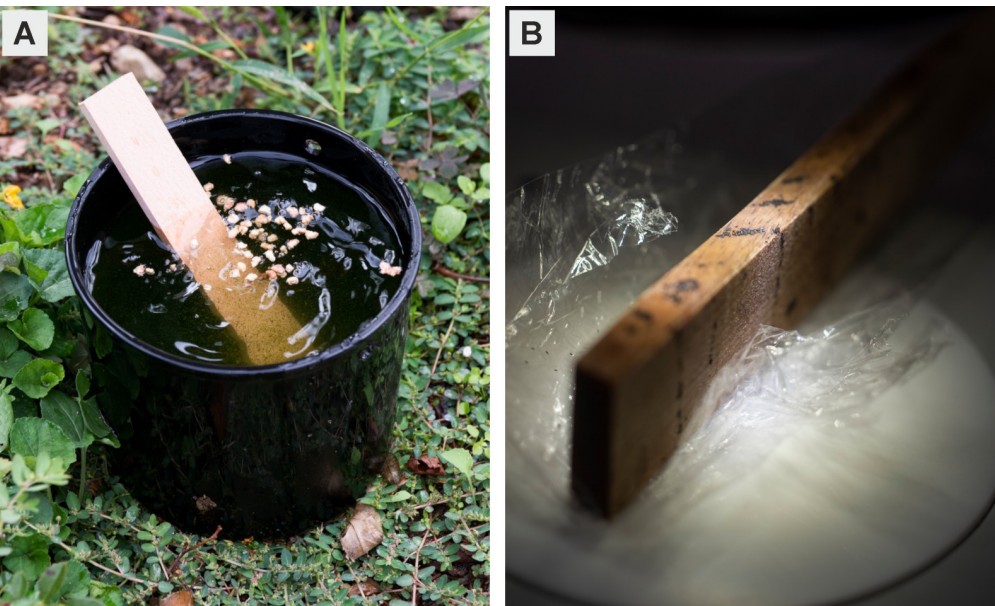

**Fig 1. Ovitrap used to collect eggs of invasive *Aedes* species. (A)** The female mosquitoes glue their eggs on the slat that is plunged into the water inside the flower pot. The slats were collected and inspected biweekly. Photo credit: Roland Schmid, Swiss TPH. **(B)** Slat with *Aedes* mosquito eggs. Photo credit: Christian Flierl, University of Basel.

At the larger motorway service areas that are accessible from both driving directions (i.e. "A1 Gunzgen" and "A1 Rose de la Broye"), the airports, the commercial harbours and the Chiasso railway station we placed six ovitraps, while only three ovitraps were set at the smaller motorway service areas that can only be accessed in one direction (Table 1). To reduce competition in attraction between the traps we set them apart by at least 50 m. At the motorway service areas we placed one trap near the entrance, one next to the main service building and one near the lorry parking, usually situated near the exit. For the ovitraps we adopted the design previously described in Flacio et al. [28]. In brief, an ovitrap consists of a black 1.5 litre plastic container, filled with about 1.2 litres of tap water (Fig 1A), into which a short wooden slat is plunged that serves as a substrate for the female *Aedes* mosquitoes to lay their eggs (Fig 1B). In each trap we added about 20–30 Vectobac granules (Valent BioSciences, Illinois, USA), containing the active *Bacillus thuringensis* var. *israelensis (Bti)*, to prevent the trap from becoming an additional breeding site. We set the traps in hidden places on the ground that were protected from direct sunlight (e.g. under vegetation or near buildings).

At each site we also aimed at placing one BG-Sentinel trap. However, as the BG-Sentinel trap requires electricity to power its fan and because we visited each site once every other week we only placed BG-Sentinel traps at sites where we had direct access to a permanent power supply (Table 1). Each BG-Sentinel trap was fitted with a BG-Lure (Biogents, Regensburg, Germany) as an attractant. In addition, we equipped six BG-Sentinel traps with a cylinder that released $CO_2$ at a constant flow rate of 175 ml per minute (Table 1).

## Species identification

Upon removing from the ovitraps, we wrapped each slat in cling film and stored the labelled and packaged slats at room temperature until they were inspected under a stereo microscope for the presence of *Aedes* eggs. Where present, we distinguished between the indigenous *Ae. geniculatus* and potentially invasive *Aedes* species. While *Ae. geniculatus* can be

unambiguously identified morphologically, the eggs of invasive species are hard to identify to species level. Therefore, we first counted the eggs of invasive *Aedes* species and then took a subsample for molecular identification. For the molecular identification, we measured protein profiles using matrix-assisted laser desorption/ionization time-of-flight mass spectrometry (MALDI-TOF MS) with an AXIMA Performance spectrometer (Shimadzu, Kyota, Japan) following the protocol of Schaffner et al. [48]. Then we sent the resulting spectra to Mabritec AG (Riehen, Switzerland) to compare them against the company's validated mosquito species reference data base.

For the identification of the adult mosquitoes, we kept the specimens at -20˚C until morphological identification under a stereo microscope [49,50]. While we identified any mosquito potentially being an invasive *Aedes* species, all other specimens were not further analysed, with the exception of the collections in 2014 when we identified all specimens to species or species complex level. In some instances, the adult mosquitoes could not be identified morphologically because they were in a bad condition or members of closely related species that could morphologically not be distinguished (e.g. *Culex pipiens/torrentium*). In these cases, we identified the specimens also with MALDI-TOF MS following a similar protocol as used for the eggs but adapted for adults [51] and sent them to Mabritec AG.

## Data analysis

Data were captured in an Access 2016 database (Microsoft, Washington, USA), then exported as separate text files for eggs and adults and finally imported into the open source software package R version 3.5.1 [52] for statistical analysis. The analysis comprised of descriptive summary statistics of the presence-absence and the numbers of eggs and adults of invasive *Aedes* species. For the ovitraps we assumed that all the eggs not identified morphologically to species level were the same species as those eggs identified by MALDI-TOF MS from the same slat. In a few cases we had more than one species on the same slat and, as we did not measure every single egg, for simplicity we accounted the total number of eggs to all species identified.

To investigate the relationship between the number of positive ovitraps and sampling year, we fitted generalised linear models (GLMs) with a logit link function and a binomial error distribution. Similarly, we modelled the number of eggs in the positive ovitraps as a function of year using GLMs with a log link function and a negative binomial error distribution in the R package "MASS" [53]. The level of significance was set at $\alpha = 0.05$.

We plotted the graphs with the R package "ggplot" [54], while the maps were drawn with ArcGIS Desktop 10.6.1 (Environmental Systems Research Institute, Inc., California, USA).

## Results

In total, we placed 5,294 wooden slats across 154 ovitraps over six years. The number of slats was slightly lower than theoretically possible (i.e. 5,544) because we could not set all traps in each year and trapping round. For example, ongoing construction work temporarily restricted access to some of the sites. From the wooden slats placed in the field, 13.1% were lost; either the traps or the slats were missing, or the traps have become dysfunctional (e.g. being tipped over, damaged or filled with rubbish). From the remaining slats 31.5% were positive. Among the positive slats (n = 1,448), 56.7% had *Ae. japonicus* eggs, 19.5% *Ae. albopictus* eggs, 1.9% *Ae. koreicus* eggs and 1.9% *Ae. geniculatus*. In 24.4% of the positive slats we had eggs that could not be identified to species level but were most likely one of the three invasive *Aedes* species. On a few occasions we also had slats with more than one *Aedes* species (4.4%). The raw data from the ovitraps are provided in S1 Data.

As with the ovitraps, we could not set all the 24 BG-Sentinel traps in Table 1 throughout the entire study and the number of traps in a year varied between 19 (2018) and 23 (2014). In contrast to the ovitraps, we could, however, reclaim and analyse all catch bags from the BG-Sentinel traps that were placed. Per trap we caught a maximum of five *Ae. albopictus* and a maximum of 10 *Ae. japonicus* adults. We had no *Ae. koreicus* in the BG-Sentinel traps. The raw data from the BG-Sentinel traps are provided in S2 Data.

Since 2013, the number of wooden slats that had eggs of an invasive *Aedes* species has increased continuously from one year to the next (odds ratio, OR = 1.23, 95% confidence interval, 95% CI = 1.19–1.28; Fig 2A). The overall annual increase in the proportion of positive traps was primarily due to a rise in the number of ovitraps with *Ae. japonicus* eggs (OR = 1.22,

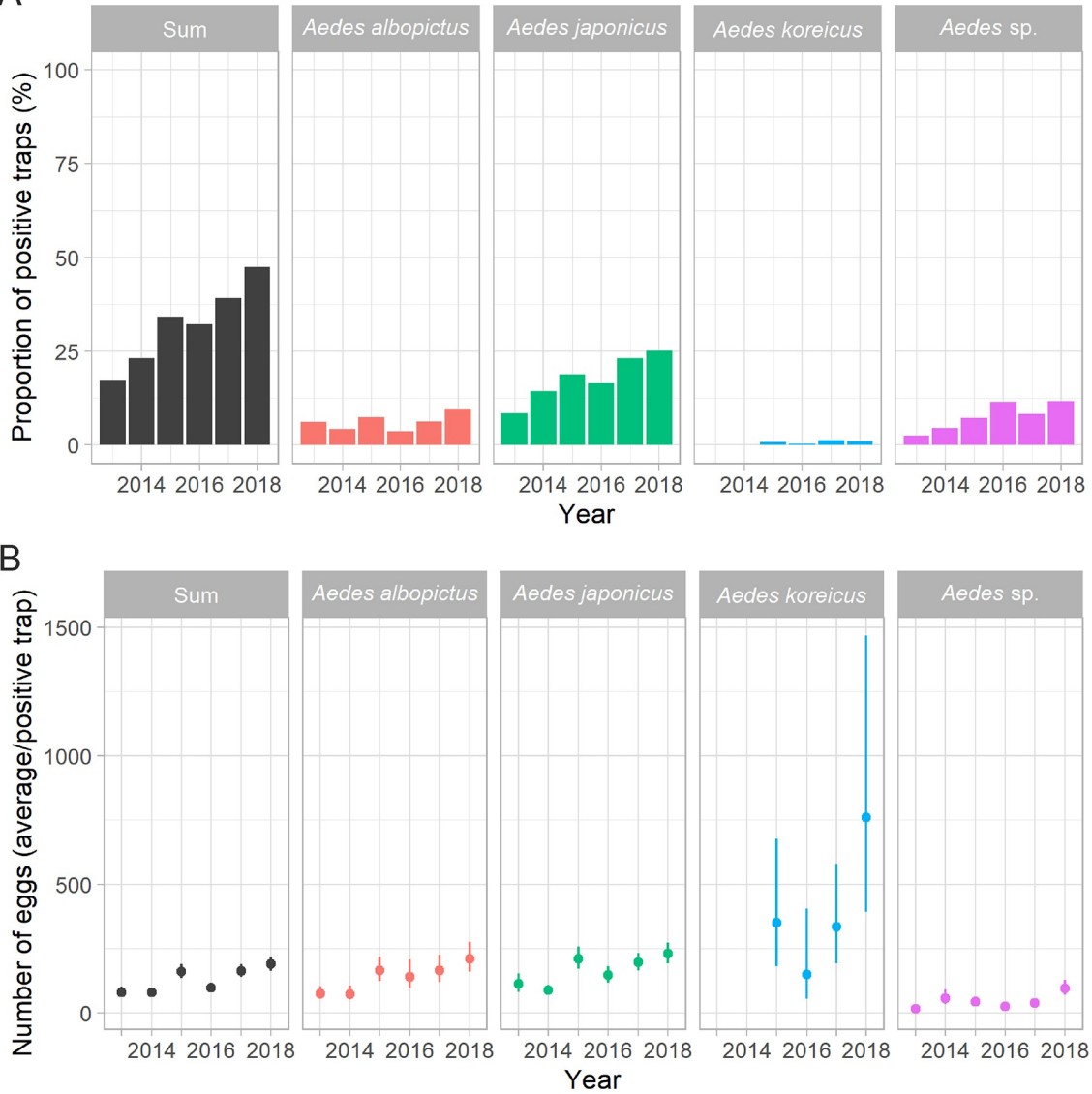

**Fig 2. Trends in the number of positive ovitraps and egg counts between 2013 and 2018. (A)** Proportion of ovitraps (i.e. slats) that had eggs. **(B)** Average number of eggs per trap among the positive traps (i.e. slats). Traps that had not been set throughout the entire study were removed from the analysis. The error bars indicate the 95% confidence intervals (95% CI) around the average estimated using the generalised linear models. The large 95% CI's for *Aedes koreicus* are due to the low number of positive traps. "*Aedes* sp." denotes all counts associated with slats where the eggs were invasive *Aedes* species but could not be identified to species level.

95% CI = 1.16–1.27). Nevertheless, the proportion of ovitraps with *Ae. albopictus* eggs has also slightly increased over the years (OR = 1.1, 95% CI = 1.02–1.18). It is also conceivable that the increased number of positive traps with unidentified eggs would, if known, further contribute to a rise in one or the other *Aedes* species. As an illustration, in 2013, the proportion of positive traps has been 17% (n = 710), while it rose to 47.5% (n = 750) in 2018, with an individual increase from 6.1% to 9.8% for *Ae. albopictus* and an almost three-fold increase from 8.5% to 25.2% for *Ae. japonicus*. In parallel to the number of positive traps the average egg count in the positive ovitraps has also increased between consecutive years (OR = 1.18, 95% CI = 1.13–1.23; Fig 2B). The ORs for *Ae. japonicus* and *Ae. albopictus* were 1.17 (95% CI = 1.11–1.23) and 1.22 (95% CI = 1.13–1.31), respectively. Since 2015, a small number of traps was sporadically also positive for *Ae. koreicus* (Fig 2).

In the collections from 2014, in addition to the invasive *Aedes* species, we also identified all other mosquito specimens to the lowest possible taxonomic level. We caught 7,424 specimens and identified 10 mosquito taxa, including six *Aedes*, one *Anopheles* and three *Culex* taxa (Table 2). The most dominant taxon (97.1%) was the sibling species *Culex pipiens/torrentium*. *Culex pipiens/torrentium* was not only the most numerous but also the most widespread taxon being present across all BG-Sentinel traps with the exception of one single site (i.e. 22 out of 23). However, the BG-Sentinel trap placed at that site had no mosquitoes caught at all. A subset of 25 individuals from five different sites, one south ("Bellinzona-Sud") and three north ("Innenhof Swiss TPH", "A2 Neuenkirch-Nord" and "A1 Thurauen-Nord") of the Alps, measured with MALDI-TOF MS revealed that they were all *Cx. pipiens* s.s. Other more frequent taxa were *Cx. hortensis*, *Anopheles plumbeus* and *Ae. japonicus* (Table 2).

*Aedes albopictus* was mainly introduced along the national motorway A2 (i.e. European route E35) from south to north as highlighted by the increased frequency of both positive ovitraps (Fig 3A) and positive BG-Sentinel traps (Fig 3B). The frequency of ovitraps also indicates a similar, though less dominant role, for the alternative south-north route over the San Bernardino (Fig 3A). In contrast, both the ovitraps (Fig 4A) and the BG-sentinel traps (Fig 4B) show a concentration of *Ae. japonicus* around the initially described distribution area in the Canton of Aargau from 2008 [36]. As we go further away from this area the frequencies are decreasing. For both *Ae. albopictus* and *Ae. japonicus* the traps in the West were negative throughout the study period (Figs 3 and 4). When comparing side-by-side the situation in terms of the presence of *Ae. albopictus* and *Ae. japonicus* between 2013 and 2018, the picture above becomes

**Table 2. Mosquito specimens sampled in the BG-Sentinel traps in 2014.**

| Species | Sites | Morphology | MALDI-TOF MS | Total |
|---|---|---|---|---|
| *Culex pipiens/torrentium* | 22 | 7188 | 25* | 7213 |
| *Culex hortensis* | 11 | 74 | 14 | 88 |
| *Anopheles plumbeus* | 7 | 49 | 9 | 58 |
| *Aedes japonicus* | 12 | 2 | 39 | 41 |
| *Aedes albopictus* | 2 | 0 | 7 | 7 |
| *Aedes vexans* | 2 | 0 | 7 | 7 |
| *Aedes geniculatus* | 2 | 0 | 4 | 4 |
| *Aedes cinereus/geminus* | 2 | 3 | 0 | 3 |
| *Aedes caspius* | 1 | 0 | 2 | 2 |
| *Culex territans* | 1 | 0 | 1 | 1 |
| Total | - | 7316 | 108 | 7424 |

*All 25 individuals measured by MALDI-TOF MS were *Culex pipiens* s.s.

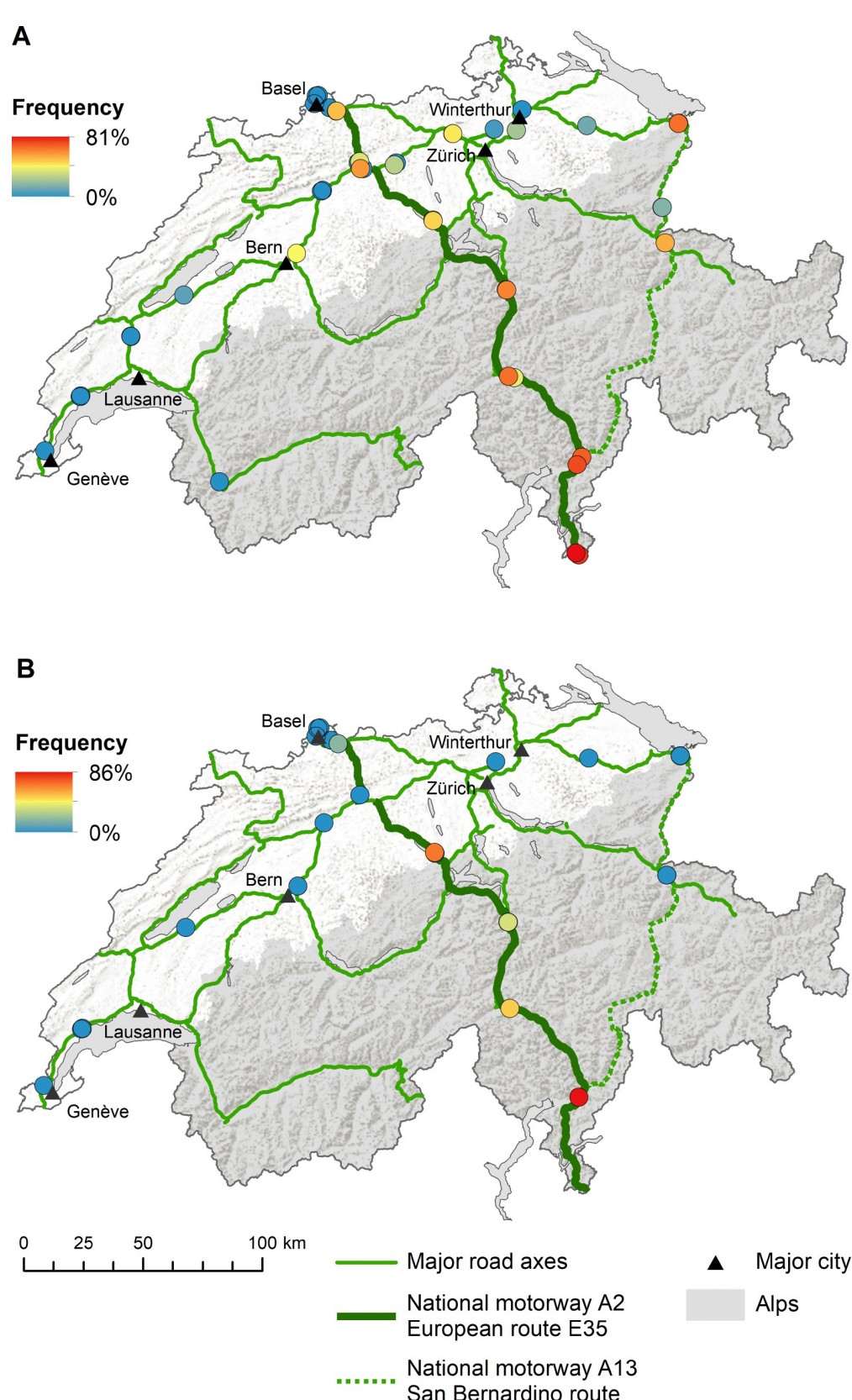

**Fig 3. Frequency of *Aedes albopictus* positive sites along the Swiss main traffic axes.** (**A**) Map showing the frequency of positive ovitraps per site. Each dot represents a site while the colour indicates how often the site was positive for *Ae. albopictus* between 2013 and 2018. Missing slats were excluded from the analysis. (**B**) Map showing the frequency of positive BG-Sentinel traps per site. Each dot represents a site while the colour indicates how often the site was positive for *Ae. albopictus* between 2014 and 2018. Missing BG-Sentinel traps were excluded from the analysis. Map source: Swiss Federal Office of Topography (swisstopo); Swiss Federal Statistical Office, Section Geoinformation (GEOSTAT) and Swiss Federal Office for Spatial Development (ARE).

even more apparent (Fig 5). *Aedes albopictus* was primarily present in the Canton of Ticino and occasionally introduced to the North in 2013, while being more frequently introduced along the south-north axis in 2018. *Aedes japonicus* was almost exclusively present in the

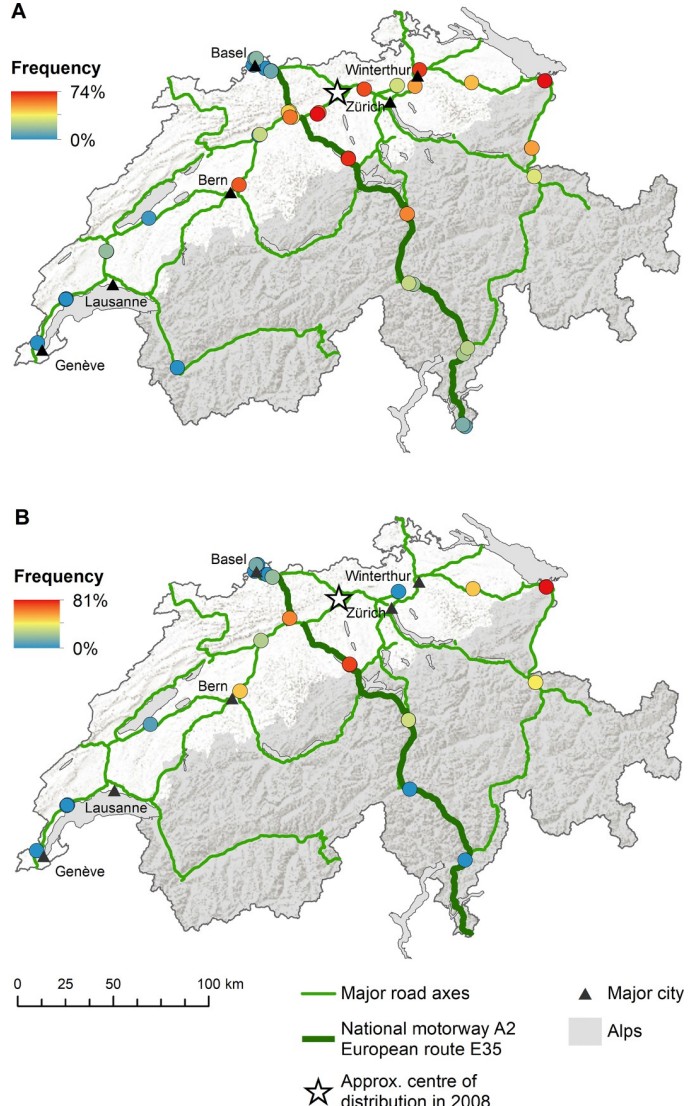

**Fig 4. Frequency of *Aedes japonicus* positive sites along the Swiss main traffic axes.** (**A**) Map showing the frequency of positive ovitraps per site. Each dot represents a site while the colour indicates how often the site was positive for *Ae. japonicus* between 2013 and 2018. Missing slats were excluded from the analysis. (**B**) Map showing the frequency of positive BG-Sentinel traps per site. Each dot represents a site while the colour indicates how often the site was positive for *Ae. japonicus* between 2014 and 2018. Missing BG-Sentinel traps were excluded from the analysis. Map source: Swiss Federal Office of Topography (swisstopo); Swiss Federal Statistical Office, Section Geoinformation (GEOSTAT) and Swiss Federal Office for Spatial Development (ARE).

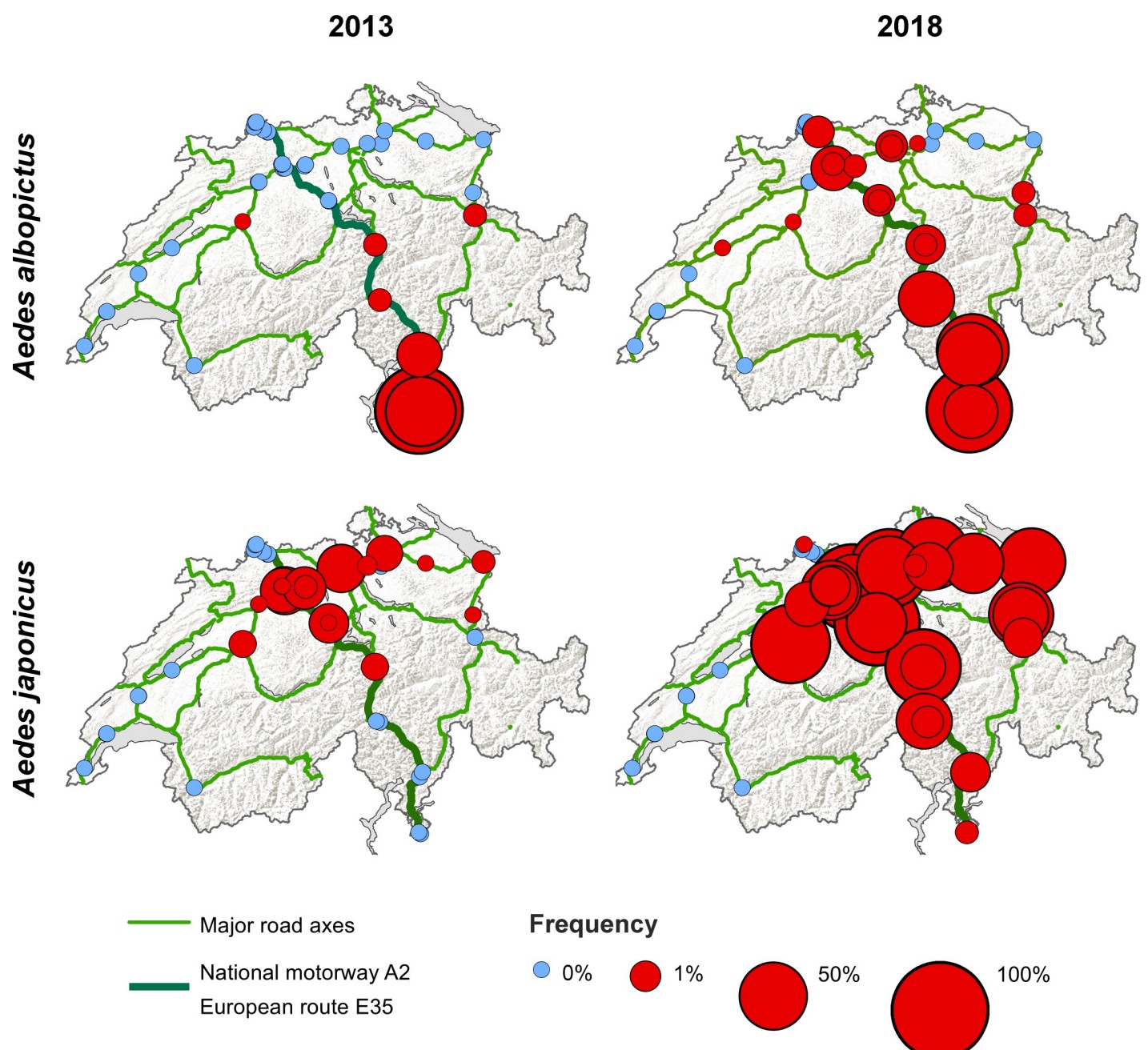

**Fig 5. Distribution and frequency of *Aedes albopictus* and *Ae. japonicus* along main Swiss traffic axes in 2013 and 2018.** Each circle represents a site while the colour and size of the circle indicates the frequency of positive ovitraps at that site. Missing slats were excluded from the analysis. Map source: Swiss Federal Office of Topography (swisstopo) and Swiss Federal Statistical Office, Section Geoinformation (GEOSTAT).

midlands in 2013 and showed a wider distribution in 2018 when it was also found in the Canton of Ticino. In line with the increased proportion of positive ovitraps and egg numbers (Fig 2), the frequency of positive ovitraps in 2018 is a lot higher than it still was in 2013 (Fig 5). Intriguingly, we found both *Ae. albopictus* and *Ae. japonicus* at the highest site at an altitude of 1,064 m (i.e. A2 San Gottardo-Sud Stalvedro) as well as at the lowest site at 227 m (i.e. A2 Bellinzona-Sud), yet neither species was present in the traps set in the West of Switzerland.

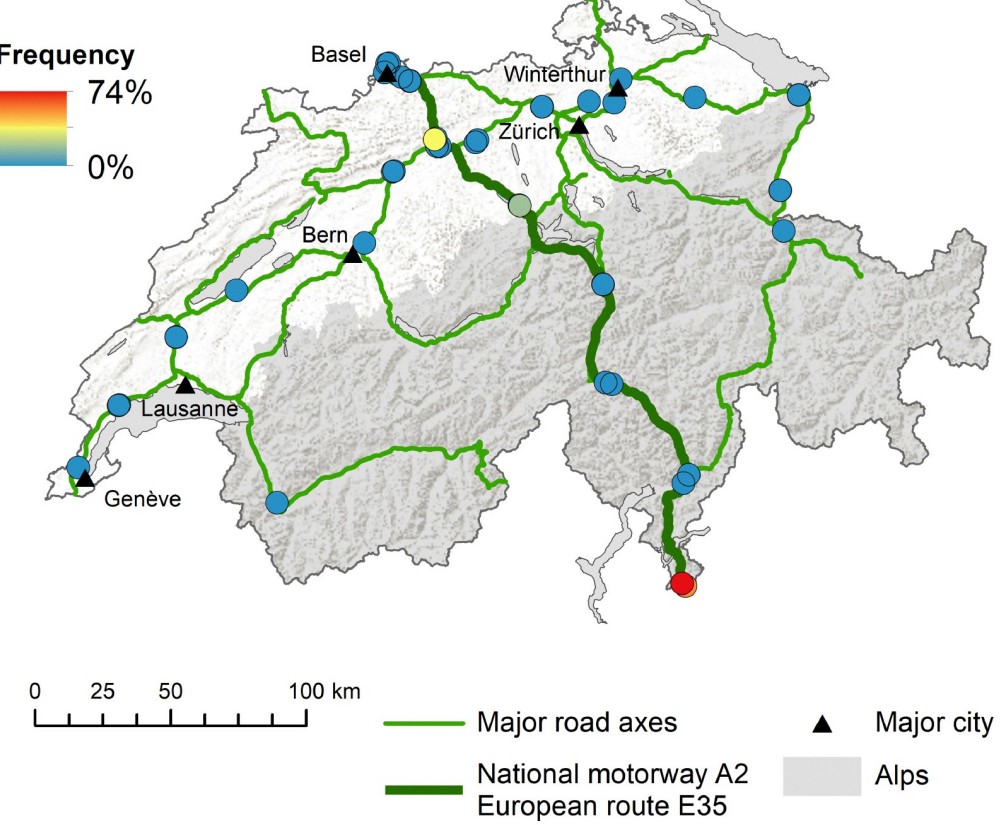

**Fig 6. Frequency of *Aedes koreicus* positive sites along the Swiss main traffic axes.** Each dot represents a site while the colour indicates the frequency of positive ovitraps per site between 2013 and 2018. Missing slats were excluded from the analysis. Map source: Swiss Federal Office of Topography (swisstopo); Swiss Federal Statistical Office, Section Geoinformation (GEOSTAT) and Swiss Federal Office for Spatial Development (ARE).

In addition to *Ae. albopictus* and *Ae. japonicus* we have also detected *Ae. koreicus* in ovitraps placed in the Canton of Ticino at the motorway service area "A2 Coldrerio" since 2015 and twice at the railway station in Chiasso in 2015 and 2017. In 2015 two sites north of the Alps, on the motorway service areas "A2 Teufengraben" and "A2 Neuenkirch-Süd", were also positive for *Ae. koreicus* (Fig 6). However, we did not find *Ae. koreicus* adults in the BG-sentinel traps.

## Discussion

The aim of this study was to monitor the presence and frequency of invasive *Aedes* mosquitoes at potential points of entry in Switzerland with a focus on motorway service areas. We found that the frequency of positive ovitraps has markedly increased between 2013 and 2018, mainly because of a surge in *Ae. japonicus*. While *Ae. albopictus* is consistently introduced along the national motorway A2 from south to north, the pattern for *Ae. japonicus* implies a more active, radial range expansion. Since 2015 we have also been recording the presence of *Ae. koreicus*, both south and north of the Alps.

Before our study, *Ae. albopictus* was already widespread in the southern parts of the Canton of Ticino following its introduction from Italy [28], but had not been reported from any other region in Switzerland north of the Alps. While the Alps may constitute a natural barrier to the dispersal of mosquitoes, until 2013, Switzerland also lacked a national surveillance programme; and hence introductions might have simply remained unnoticed. Indeed, in 2007,

*Ae. albopictus* eggs had already been detected in the Upper Rhine Valley in Germany at a motorway service area on the A5 near the Swiss border [55], followed by additional reports from the same region in 2011 [56] and in 2012 [57]. The German motorway A5 is the extension of the Swiss motorway A2. Both motorways constitute sections of the European Route E35 that runs from Rome, Italy, in southern Europe to Amsterdam, the Netherlands, in the north-western part of the continent. Because Italy is regarded as the primary source for *Ae. albopictus* in Europe [25], it does not come as a surprise that, in addition to the positive sites in Germany, multiple sites in Switzerland along the A2 were also frequently positive. Likewise, the passive spread of *Ae. albopictus* from south to north would explain our repeated findings along the San Bernardino route A13, an alternative south-north passage way through Switzerland. A similar picture emerges from a study in western Austria (Tyrol) where service stations along the motorways coming from Italy were repeatedly positive [44]. Together these observations support the general notion that *Ae. albopictus* is primarily being passively spread by motorised vehicles from south to north along the main road axes, and that the Alps represent no physical barrier for its expansion in Central Europe.

In contrast to the prominent south-north pattern we do not see much evidence in our data for an *Ae. albopictus* dispersal along the west-east axis. The motorway service areas in Western Switzerland near Geneva remained negative throughout the study, despite the mosquito had already been well established in southern France before the present study and has been reported from multiple sites along the Rhone Valley as close as Geneva [58]. An explanation could be lower mosquito densities in France as compared to Italy, reduced traffic volumes between the infested areas in France and Western Switzerland when compared to the south-north circulation, fewer vehicles stopping at the motorway service areas included in our study, or a combination of these factors.

In contrast to *Ae. albopictus*, that shows a pattern consistent with passive dispersal through motorised vehicles, the distribution and frequency of sites positive for *Ae. japonicus* over the years imply a more active range expansion. In 2008, the initially colonised area was estimated at 1,400 km$^2$, covering parts of Switzerland and bordering Germany [36]. Within that original range we had six sites in the present study, including the motorway service stations "A1 Kölliken-Nord", "A1 Kölliken-Süd ", "A1 Würenlos-Nord", "A1 Würenlos-Süd", "A1 Kempthal" and "Zürich Airport". Between 2013 and 2018, these sites were frequently positive for *Ae. japonicus*, while the sites at the periphery were positive depending on the distance away from the initial distribution area. Although passive distribution via traffic is generally assumed to be the chief mode of dispersal of invasive *Aedes* mosquitoes [59], our data for *Ae. japonicus* are more consistent with the hypothesis of active dispersal being the key driver. This would not exclude that passive dispersal along motorways may still take place as the more recent findings in the Canton of Ticino imply. Interestingly, a previous study of *Ae. japonicus* found that its larvae are more frequently present in rural than urban areas as compared to *Ae. albopictus* [60], which might reflect an adaptation allowing this species to disperse more actively across a heterogeneous landscape. Though population genetic studies would be more informative to test the hypothesis of active range expansion, the circumstantial evidence from this study supports the idea of active range expansion for *Ae. japonicus*.

In addition to *Ae. japonicus* we found *Ae. koreicus* in several of our ovitraps along the south-north Gotthard route. Given the few findings in our study the picture is still somewhat unclear, but this mosquito seems to be more present in the South of the country and, like *Ae. albopictus*, has likely been passively dispersed along the south-north axis through motorised vehicles driving up the motorway A2. This is in line with its first detection at the Swiss-Italian border in 2013 [40] and with previous reports from northern Italy [61]. Moreover, *Ae. koreicus* has also been repeatedly reported from Germany [30,62,63] and more recently from Austria

[44]. Like the closely related *Ae. japonicus*, *Ae. koreicus* is a container-breeding mosquito species that is well adapted to a more temperate climate [34]; and hence we expect that this species, too, will become more widespread across Switzerland and the rest of Central Europe.

Although primarily used to trap host-seeking *Ae. albopictus*, the BG-Sentinel trap has also been found suitable for the surveillance of other adult mosquitoes in Europe [64]. With this in mind we also identified all other adult specimens that were collected in the BG-Sentinel traps to the lowest possible taxonomic level. However, we were only able to do this extra effort in 2014. In that year we identified 10 different taxa from about 40 known species in Switzerland [40, 65, 66]. Among the 10 taxa found in the BG-Sentinel traps the most dominant one was the sibling species pair *Culex pipiens/torrentium*. *Culex pipiens/torrentium* has frequently been described as the most abundant mosquito taxon in Switzerland [66, 67]. Further MALDI-TOF MS analysis of a subset of specimens taken from one site in Ticino and three sites north of the Swiss Alps identified *Cx. pipiens* s.s. only and no *Cx. torrentium*. These results are in line with a more comprehensive Swiss study by Wagner et al. [68] who found that *Cx. torrentium* shows very low relative abundances across suburban and natural areas north of the Swiss Alps. *Culex hortensis* was the second most abundant species, followed by *An. plumbeus* and *Ae. japonicus*. These species have also been observed at higher frequencies in Switzerland in previous studies [36,67].

Both the ovitraps and the BG-Sentinel traps were set during the putative peak activity period of *Ae. albopictus* from June to September, and it is possible that we might have missed introductions of invasive *Aedes* mosquitoes before or after this period. Besides, we were not able to identify every single egg to species level, so that the actual numbers of eggs per species have some uncertainties. Another point of consideration, primarily for the adult trapping, is that choices for the positioning of the BG-Sentinel traps were limited because we had to rely on the power supply from the mains to operate them. Moreover, the traps had to be well concealed from the public to avoid vandalism. Similarly, only a small subset of BG-Sentinel traps could be fitted with a $CO_2$ bottle, meaning that they likely differed in their attraction to mosquitoes. Altogether, this might have introduced a sampling bias. Also, in 2013 we had to rely on ovitraps only. Therefore, a direct comparison between the two datasets is not warranted, however, where we have higher egg densities the BG-Sentinel traps are generally in agreement with the ovitraps.

As previously documented for *Ae. albopictus* [28,69] invasive *Aedes* mosquitoes escaping from vehicles upon stopping at motorway service areas may constitute critical points of entry for the early establishment of new populations, or potentially add to existing infestations, particularly if the points of entry are close to residential areas. In 2018 an attentive citizen reported an adult *Ae. albopictus* 300 m away from the motorway A1 service area "Gunzgen-Nord"—where ovitraps were repeatedly positive—suggesting that the mosquito has either escaped from a vehicle stopping there or its offspring have made their way to the village. A recent study suggests that newly hatched adult *Ae. albopictus* mosquitoes fly several hundred metres away from their breeding site to seek hosts [70]. Therefore, unless effective control measures are implemented, several sites north of the Alps are at risk of becoming starting points for the establishment of new populations.

The regions north of the Alps are likely to be more suitable for the establishment of permanent *Ae. japonicus* rather than *Ae. albopictus* populations due to the former being better adapted to temperate climates [71,72] and because of its extended phenology [39]. Similarly, *Ae. koreicus* might also be more adapted to the colder climate north of the Alps [41]. However, while many areas might not provide ideal habitats for *Ae. albopictus* because of temperature limits during the winter months, warmer regions such as the area around the lake of Geneva, the Upper Rhine Valley, including Basel, or at the shores of Lake Constance might still be suitable enough for the establishment of locally reproducing populations as habitat suitability

models [71], microclimatic studies [73] and recent reports from Basel [74] and southern Germany suggest [75,76]. To shed more light into this question, future work should consider population genetic approaches to identify the sources of the *Ae. albopictus* specimens found north of the Alps.

From our results we conclude that *Ae. albopictus* and *Ae. koreicus* are being passively spread primarily along the European route E35, while *Ae. japonicus* is most likely expanding its range largely through active dispersal. Because of increasing introduction of invasive *Aedes* mosquitoes, control measures should be put in place to reduce the likelihood of their establishment from points of entry such as motorway service stations.

## Supporting information

**S1 Data. Original data set with egg counts for each wooden slat.** Each line corresponds to a single observation. "sitecode" = code of the sampling site; "trapid" = unique identifier of the trap; "year" = sampling year; "round" = sampling round (round 1 to round 6); "date" = date when the slat was recovered from the ovitrap; "status" = indicates whether the trap was functional or not ("1" = the trap was working ok, "0" = there was a problem with the trap); "n.eggs" = egg count; "n.maldi" = number of eggs measured with MALDI-TOF MS; "species" = mosquito species; "albopictus" = presence of *Ae. albopictus* eggs ("1" = yes, "0" = no); "japonicus" = presence of *Ae. japonicus* eggs ("1" = yes, "0" = no); "koreicus" = presence of *Ae. koreicus* eggs ("1" = yes, "0" = no); "geniculatus" = presence of *Ae. geniculatus* eggs ("1" = yes, "0" = no); "aedes" = presence of *Aedes* spp. eggs ("1" = yes, "0" = no); "site" = name of the sampling site; "sitetype" type of the sampling site ("Autobahn" = motorway service area, "Bahnhof" = railway station; "Flughafen" = airport, "Hafen" = commercial harbor, "Stadt" = city); "canton" = acronym of the Swiss canton (NUTS-3) where the sampling site was located; "long" = geographical longitude in the World Geodetic System format WGS84; "lat" = geographical latitude in the World Geodetic System format WGS84; "altitude" = altitude of the sampling site in metres. (CSV)

**S2 Data. Original data set from the BG-Sentinel traps.** Each line corresponds to a single observation. "sitecode" = code of the sampling site; "location" = name of the sampling site; "trapid" = unique identifier of the trap; "year" = sampling year; "round" = sampling round (round 1 to round 6); "CO2" = indicates whether the trap was equipped with $CO_2$ (can be "With CO2" or "Without CO2"); "long" = geographical longitude in the World Geodetic System format WGS84; "lat" = geographical latitude in the World Geodetic System format WGS84; "date.set" = date when the trap was set; "date.control" = date when the catch bag was removed from the trap; "status" = indicates whether the trap was functional or not ("ok" = the trap was working ok, "Altered" = there was a problem with the trap); "species" = mosquito species; "n.adults" = adult count (from the same species and sex); "sex" = sex; "id.method" = method of species identification ("MALDI-TOF MS" = identification with MALDI-TOF MS, "Morphology" = morphological identification); "sitetype" = type of the sampling site ("Autobahn" = motorway service area, "Flughafen" = airport, "Hafen" = commercial harbor, "Stadt" = city); "canton" = acronym of the Swiss canton (NUTS-3) where the sampling site was located; "altitude" = altitude of the sampling site in metres. (CSV)

## Acknowledgments

This work would have not been possible without the support of many pairs of hands both in the field and the laboratory. For the hard work in collecting and replacing traps, enduring

traffic jams, measuring mass spectra or spending endless hours in counting eggs we are extremely grateful to Valentina Alesi, Nikoleta Anicic, Leandro Balzarini, Anna-Paola Caminada, Alissa Cereghetti, Alison Crenna, Luca Davino, Sophie DeRespinis, Begoña Feijoó Fariña, Attila Giezendanner, Riccardo Hefti, Alida Kropf, Elian Kuhn, Diego Parrondo Montón, Natalia Rava, Laura Rieder, Andrea Tavasci and Seraina Vonzun. It was also a pleasure to work with Valentin Pflüger and Roxane Mouchet from Mabritec AG for the MALDI-TOF MS analysis. We would also like to thank Peter Lüthy for his initial encouragement to set-up a national surveillance programme for invasive mosquitoes in Switzerland. Last but not least, a big thank you to Basil Gerber and Thomas Probst from the Swiss Federal Office for the Environment for their continuous support and encouragement throughout the project.

## Author Contributions

**Conceptualization:** Pie Müller, Lukas Engeler, Tobias Suter, Mauro Tonolla, Eleonora Flacio.

**Data curation:** Pie Müller, Lukas Engeler, Tobias Suter.

**Formal analysis:** Pie Müller, Lukas Engeler, Laura Vavassori.

**Funding acquisition:** Pie Müller, Mauro Tonolla.

**Investigation:** Pie Müller, Lukas Engeler, Laura Vavassori, Tobias Suter, Valeria Guidi, Martin Gschwind.

**Methodology:** Pie Müller, Lukas Engeler, Tobias Suter.

**Project administration:** Pie Müller.

**Supervision:** Pie Müller, Mauro Tonolla, Eleonora Flacio.

**Validation:** Pie Müller, Lukas Engeler, Laura Vavassori.

**Writing – original draft:** Pie Müller, Laura Vavassori.

**Writing – review & editing:** Pie Müller, Lukas Engeler, Laura Vavassori, Tobias Suter, Valeria Guidi, Martin Gschwind, Mauro Tonolla, Eleonora Flacio.

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
