## [Decision Letter · Decision Letter 0]

1 Jul 2020

Dear Dr Mueller,

Thank you very much for submitting your manuscript "Surveillance of invasive Aedes mosquitoes along Swiss traffic axes reveals different dispersal modes for Aedes albopictus and Ae. japonicus" for consideration at PLOS Neglected Tropical Diseases. As with all papers reviewed by the journal, your manuscript was reviewed by members of the editorial board and by several independent reviewers. The reviewers appreciated the attention to an important topic. Based on the reviews, we are likely to accept this manuscript for publication, providing that you modify the manuscript according to the review recommendations. 

Sincerely,

Mariangela Bonizzoni

Associate Editor

Amy Morrison

Deputy Editor

Reviewer's Responses to Questions

**Key Review Criteria Required for Acceptance?**

**Methods**

-Are the objectives of the study clearly articulated with a clear testable hypothesis stated?

-Is the study design appropriate to address the stated objectives?

-Is the population clearly described and appropriate for the hypothesis being tested?

-Is the sample size sufficient to ensure adequate power to address the hypothesis being tested?

-Were correct statistical analysis used to support conclusions?

-Are there concerns about ethical or regulatory requirements being met?

Reviewer #1: (No Response)

Reviewer #2: Methods are appropriate and statistical analyses are sound.

**Results**

-Does the analysis presented match the analysis plan?

-Are the results clearly and completely presented?

-Are the figures (Tables, Images) of sufficient quality for clarity?

Reviewer #1: (No Response)

Reviewer #2: Results are in line with the objectives of the study and completely presented. Figures and tables are enough

**Conclusions**

-Are the conclusions supported by the data presented?

-Are the limitations of analysis clearly described?

-Do the authors discuss how these data can be helpful to advance our understanding of the topic under study?

-Is public health relevance addressed?

Reviewer #1: (No Response)

Reviewer #2: Conclusions are supported by the results.

**Editorial and Data Presentation Modifications?**

Reviewer #1: (No Response)

Reviewer #2: (No Response)

**Summary and General Comments**

Reviewer #1: The authors monitored alien Aedes mosquitoes along Swiss traffic axes with ovitraps (egg collection) and BG Sentinel traps (adults) – from June-September 2013-2018. Aedes albopictus, Ae. japonicus and Ae. koreicus were documented. Based on their results Ae. albopictus and Ae. koreicus are passively spread while Ae. japonicus is spreading actively.

This manuscript is of excellent quality and provides important information about the alien Aedes mosquito situation in Switzerland. It is also of high relevance for public health.

Some minor modifications are recommended (and I apologize that I mention our studies but it would be of interest to compare and discuss)

Line 20: You could also include Zika here

Line 38: These Aedes mosquitoes are mainly of relevance for human medicine 

entire manuscript: Please unify – Ae. japonicus or Ae. japonicus japonicus

entire manuscript: use of invasive – We all know that Ae. albopictus is an invasive species - however, the status of invasiveness in the Central and Eastern Alpine regions remain unclear (outcompetition of indigenous mosquitoes beside establishment). Therefore, it is recommended to use potential invasive or alien

Fuehrer et al. (2020 – PLoS NTDs) published a study about these mosquitoes at high ways in the Eastern Alps region of Austria (Tyrol) a few days ago. It is recommended to compare results.

Line 118: Please provide information which BG-Sentinel generation was used 

Line 228: New paragraph after koreicus

Change Culex pipiens s.s. to Culex pipiens complex

General question: Did you observe any altitude differences? We observed Ae. japonicus in mountainous areas above 850 m (Schoener et al. 2019)

Line 383: Change , to .

Line 407: Cx. torrentium abundance – change to “… north of the Swiss Alps” or modify – The statement is not true for the entire northern to the Alps regions (e.g. Zittra et al. 2016; “…. Under-representation of Cx. torrentium in carbon dioxide baited traps is commonly observed”).

Reviewer #2: The manuscript of Muller et al. reports a 6-years survey aimed at recording the potential presence of invasive mosquitoes (Aedes albopictus, Aedes koreicus and Aedes j. japonicus) in Switzerland at selected sites. The study is straightforward and properly conducted. It is evident the huge work done to obtain this set of data, which demonstrated its usefulness in detecting all the three invasive species, allowing also to hypothesize their spreading strategies through the country. I have only a few minor comments:

Lines 93-96: the introduction seems to me unbalanced in describing dispersal and invasiveness of Aedes albopictus while no information is given for the other two Aedes on which the manuscript is focused. I suggest a few more words about distribution and potential invasiveness of Ae. koreicus and Ae. japonicus in order to give a more complete picture of the potential risk of invasion in Switzerland.

Table 1: please explain in caption the meaning of the asterisk next to the number in BG sentinel column (I guess to highlight the traps with the presence of CO2 supply).

Lines 245-257: it could be interesting to know if the BG sentinel traps baited with BG lure+CO2 collected differently to the others with just the BG lure and if this affected data analysis in terms of detection of invasive species.

PLOS authors have the option to publish the peer review history of their article (what does this mean?). If published, this will include your full peer review and any attached files.

Reviewer #1: Yes: Hans-Peter Fuehrer

Reviewer #2: No
---

## [Editor Report · Decision Letter 1]

12 Aug 2020

Dear Dr Mueller,

We are pleased to inform you that your manuscript 'Surveillance of invasive Aedes mosquitoes along Swiss traffic axes reveals different dispersal modes for Aedes albopictus and Ae. japonicus' has been provisionally accepted for publication in PLOS Neglected Tropical Diseases.

Best regards,

Mariangela Bonizzoni

Associate Editor

Amy Morrison

Deputy Editor

---

## [Editor Report · Acceptance letter]

17 Sep 2020

Dear Dr Mueller,

We are delighted to inform you that your manuscript, "Surveillance of invasive *Aedes* mosquitoes along Swiss traffic axes reveals different dispersal modes for *Aedes albopictus* and *Ae. japonicus*," has been formally accepted for publication in PLOS Neglected Tropical Diseases.

Best regards,

Shaden Kamhawi

co-Editor-in-Chief

Paul Brindley

co-Editor-in-Chief
